# Towards Crowdsourced Training of Large Neural Networks using Decentralized Mixture-of-Experts

**Max Ryabinin**[*]
Yandex
National Research University
Higher School of Economics
mryabinin@hse.ru

**Anton Gusev**
Independent
uartman@mail.ru

## Abstract

Many recent breakthroughs in deep learning were achieved by training increasingly larger models on massive datasets. However, training such models can be prohibitively expensive. For instance, the cluster used to train GPT-3 costs over \$250 million[2]. As a result, most researchers cannot afford to train state of the art models and contribute to their development. Hypothetically, a researcher could crowdsource the training of large neural networks with thousands of regular PCs provided by volunteers. The raw computing power of a hundred thousand \$2500 desktops dwarfs that of a \$250M server pod, but one cannot utilize that power efficiently with conventional distributed training methods. In this work, we propose Learning@home: a novel neural network training paradigm designed to handle large amounts of poorly connected participants. We analyze the performance, reliability, and architectural constraints of this paradigm and compare it against existing distributed training techniques.

## 1 Introduction

Our investigation begins with a thought experiment. Imagine a deep neural network with capacity 1000 times greater than today's most powerful architectures: for example, a language model trained on all digitally available texts or a generative model for all images ever uploaded to the Internet. How can we train such a model?

Viewed from a historical perspective, the 1000-fold increase in capacity is not unrealistic. Over the past decade, the deep learning community has made remarkable progress by training large models on abundant data, and the scale of those models keeps growing. Since the advent of the ImageNet challenge [1] with 1.3M labeled images, the typical size of convolutional neural networks increased from a few megabytes to hundreds of megabytes [2, 3, 4]. Recent studies report even larger models for datasets with hundreds of millions of images [5, 6].

Another trend from natural language processing is to train large Transformer-like language models [7, 8, 9]. The data for this task is nearly unlimited, allowing researchers to train models with tens or even hundreds of gigabytes of parameters [10, 11, 12, 13]. While we may not need the 1000-fold increase at the moment, planning for it will prepare us for the next big leap in model capacity.

To be specific, let us focus on training large Transformer networks for the language modeling task. At the time of writing, the largest conventional model for that task is GPT-3 with 175 billion parameters. Scaling it up 1000 times gives us 175 trillion; depending on whether you use single or half-precision, this requires 300–600 terabytes of memory just to store the model. No modern mass-produced hardware accelerator is up to such task. Even high-end servers with 16x V100 accelerators can store only 0.15% of that model in combined GPU memory, let alone train it.

---

[*]Corresponding author.

[2]A conservative estimate based on https://blogs.microsoft.com/ai/openai-azure-supercomputer

The dominant way of growing neural network size has so far been to scale up: deploy more powerful computational accelerators in specialized tightly interconnected clusters. However, this approach will only work up to a point. Models such as T-NLG [13] and Megatron-LM [11] were already trained on DGX-SuperPOD — a supercomputer with hundreds of Tesla V100 GPUs spread over tens of servers. As for GPT-3 [10], a single *training run* was estimated to cost 4.6 – 12 million dollars [14, 15].

Even today, the need for costly hardware weighs heavily on the research community. Most researchers cannot contribute to the development of large neural networks because conducting the necessary experiments would be too expensive for them. If we continue to increase the model size by scaling up, eventually the only labs that can conduct competitive research will be those with massive budgets.

However, there is another solution: to scale out. Instead of using a supercomputer, researchers could crowdsource the computation from volunteers with regular PCs. This paradigm is known as volunteer computing and was successfully applied to solve problems in biology [16], high energy physics [17] and other subject areas. While a single volunteer PC may be slow and unreliable, the combined floating-point performance of such projects is on par with largest supercomputers [18].

The main challenge of volunteer computing is how to utilize this performance. Unlike server pods, consumer-grade PCs communicate over the Internet, which is significantly slower, especially in terms of latency. They are also more prone to failures as they lack many reliability features of their server-grade counterparts. Therefore, volunteer computing was traditionally used for tasks that have high computation to communication ratio and can recover from individual node failures.

Unfortunately, existing paradigms of distributed training require nodes to continuously transfer large amounts of intermediate data [19, 20], making them unsuitable for volunteer computing. In this work, we take a different approach. Instead of adopting the existing distributed training strategies, we identify the advantages of volunteer computing and design a new strategy that capitalizes on them.

We summarize the contributions of our paper as follows:

- We propose Decentralized Mixture of Experts (DMoE) — a layer designed for training with vast amounts of unreliable consumer-grade hardware;

- We describe a framework for training large neural networks composed of DMoE layers;

- We confirm the efficiency and reliability of this approach using formal guarantees and experiments;

- The PyTorch source code that can be used to reproduce our results is available online[3].

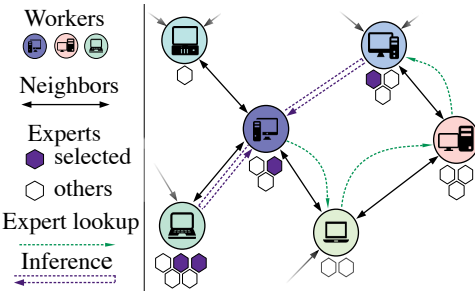

Figure 1: High-level scheme of Decentralized Mixture of Experts. See Section 3 for details.

## 2 Related work

### 2.1 Volunteer computing

Using volunteer hardware has long been a viable alternative to high-performance computing. Since the development of BOINC [21] research organizations with sufficient public outreach have been able to run massive scientific computations on devices provided by volunteers. Successful projects such as Folding@home can have over $10^5$ active participants, rivaling the floating-point performance of world's fastest supercomputers[4]. In fact, Folding@home was the first "supercomputer" to reach both 1 and 10 petaflops milestones [22].

However, unlike traditional HPC, the volunteer nature of these projects imposes some additional limitations. First, the majority of volunteers are only available part-time. For instance, a participant can provide an office workstation that only contributes compute outside of business hours. Second, volunteer hardware is heterogeneous: different nodes may have different performance, memory limits, and even operating systems. Finally, participants usually communicate over the Internet, which is 2–3 orders of magnitude slower than typical HPC connections. As a result, both compute nodes and communication channels are not nearly as reliable as in traditional supercomputers.

Due to the limitations mentioned above, volunteer computing works best for tasks that can be split into many independent chunks. A single Folding@home task is to run a physical simulation of a protein for a specified number of frames. Together, volunteers can perform hundreds of thousands of concurrent tasks and only need to communicate with the server to submit their results. Other projects like SETI@home and Einstein@home follow a similar pattern.

Based on the existing volunteer computing projects, we formulate the following usage scenario:

- **Large pool of weak computers:** the infrastructure consists of $10^3 \sim 10^6$ heterogeneous PCs[5];

- **Communication:** nodes communicate with speed and reliability of a home internet connection[6];

- **Frequent node failures:** a compute node may fail to process a task for a variety of reasons. We expect 5–20% of computers to have at least one failure a day under normal operating conditions.

## 2.2 Distributed training

To analyze the existing distributed training approaches from the perspective of volunteer computing, we broadly divide them into several categories.

**Synchronous data parallel training** [25]. Each worker stores a copy of model parameters, computing gradients for a fraction of the training batch. The gradients are then averaged across workers and applied to the model, making up the same update on all machines. Due to its simplicity and scalability, this method has been widely used to reduce the training time of large neural networks to the order of minutes [26, 27].

However, with low-end or midrange hardware it is not always possible to store the entire model on each worker. In addition, gradient communication, even when overlapped with computation, requires a high-speed connection between all participants, often faster than hundreds of megabytes per second, which is unrealistic when considering typical household Internet connections.

**Asynchronous training** [28, 29] usually involves a single parameter server and multiple compute nodes fetching the latest parameters, processing batches, and submitting updates back to the server. This technique improves worker throughput, but this improvement comes at a cost. If several workers submit simultaneous updates, they might get applied in an arbitrary order, which leads to the issue of *stale gradients* [30] and possibly hinders model convergence.

**Model parallel training.** Each node stores a fraction of model layers, each training batch is processed by all nodes in a sequential order determined by the layer distribution scheme. The training batch can be divided into several micro-batches and processed in a pipeline fashion, significantly increasing hardware utilization [4, 31, 32, 33].

Unlike the two previous paradigms, this method allows training models that exceed the memory limit of any individual worker. Notable examples of successful model parallel training for large neural networks are [4] and [11], yet these systems also have a high-speed network between workers. On top of that, model parallelism is highly vulnerable to node and network failures: if a single worker in a chain turns off or stops sending outputs, the training stops entirely.

It is possible to combine data and model parallelism to mitigate the outlined issues to some degree, but the requirement for fast worker interconnect holds even in that case. In light of this, the method we design has to maintain high throughput even in the presence of slow and unreliable network connections, possibly sacrificing the latency (time to process a given batch) as a necessary tradeoff.

This constraint may be justified by the following observation: the wall-clock training time of a neural network (with model and optimizer fixed) mostly depends on how many batches it processes per second. As we show in Section 4.2, the effect of stale gradients can be mitigated with the right architecture. We summarize the desired properties in Table 1.

**Federated learning.** The problem of utilizing large quantities of consumer devices for training a single model has also been discussed within the context of data-private learning. Federated learning [34] attempts to mitigate the issue by keeping the data on devices, training a local version of the model, and sending only the parameter updates. These updates are encrypted so that the server can only decrypt their average across several devices.

Table 1: Comparison of distributed training schemes in the volunteer computing context. "Desired" denotes the algorithm with properties that would be beneficial for this setting. "Only workers" means that the system has central components that are not fault-tolerant.

| Training method | Model size limit | Training throughput | Scalability | Fault tolerance | Worker hot-join | Network Bandwidth | Latency |
|---|---|---|---|---|---|---|---|
| Data parallel | Worker | **High** | Medium | **Full** | **Yes** | **High** | Low |
| Asynchronous | Worker | **High** | **High** | Only workers | **Yes** | Medium | **Any** |
| Model parallel | **System** | Medium | Low | No | No | High | Low |
| Federated | Worker | Low | **High** | Only workers | **Yes** | **Low** | **Any** |
| Desired | **System** | **High** | **High** | **Full** | **Yes** | **Low** | **Any** |

Unsurprisingly, federated learning sacrifices performance for privacy. Secure aggregation procedures [35] require multiple workers to communicate and scale quadratically with their number. These properties hardly align with the scenario from Section 2.1, making federated learning a poor fit for jointly training large models.

**Deep learning with volunteer computing.** To the best of our knowledge, there are three projects that use volunteer computing for training neural networks. The first work [36] leverages volunteer resources for evaluation of CNN architectures generated by evolution algorithms; each model is trained on a single device. The second study [37] relies on standard asynchronous training and is therefore inapplicable to models that do not fit into a single consumer-grade GPU. Moreover, the architecture described in that study is only partially decentralized, relying on a centralized parameter server that communicates with all nodes. Lastly, the project known as Leela Chess Zero [38], relies on volunteer hardware to play massive amounts of chess games for generating self-play data used in reinforcement learning. However, the model itself is trained on a single central server.

Our primary insight from this section is that existing methods for training general large neural networks do not fit well into the volunteer computing scenario. However, there is a subclass of deep learning architectures which is much better suited for this task.

### 2.3 Mixture-of-Experts

Mixture-of-Experts (MoE) was first proposed almost three decades ago as a method to train multiple neural networks ("experts") for a common task [39]. The intent is for each expert to specialize in making predictions for a small subset of data. Presented with an input, MoE first determines which experts are best suited to process that input using a separate *gating function*. Then it applies the chosen experts and aggregates their outputs into the final prediction. This work has sparked many follow-ups that reveal different MoE structures [40, 41, 42, 43] and individual expert types [44, 45].

A subsequent study [46] demonstrates that Mixture-of-Experts can be used as a layer within larger neural networks and trained jointly by backpropagation. Depending on the task, individual experts can utilize convolutional, recurrent, or other specialized layers. Such MoE can have a large number of experts, but it only needs to compute a few of them to process any given input.

Shazeer et al. [47] (and later [48]) brought that idea to the extreme by training "outrageously" large mixtures with thousands of experts. The drastic increase in capacity allows authors to achieve superior performance in large-scale machine translation and language modeling. The paper also addresses problems that arise with increased mixture size. When trained naïvely, the gating function learns to use a small fraction of available experts for all inputs, not taking full advantage of the available capacity. The authors alleviate this issue by adding a regularization term that promotes "load-balancing" across all experts.

However, scaling this approach from thousands to millions of experts reveals additional problems in the design of a gating function. In order to choose the most appropriate experts for the task, MoE predicts a "priority" value for each expert and selects the ones with the highest priority. As the number of experts approaches millions, such a gating function itself becomes computationally intractable, especially in our decentralized setting.

A popular solution to this problem is to structure the set of experts in a search-friendly way. For instance, Hierarchical Mixture-of-Experts [40] organizes experts in a tree-like structure. Selecting the best experts is then reduced to a beam search over this tree, which scales logarithmically in the

number of experts. More recent study by Lample et al. [49] explores this idea at scale by organizing over a million keys in a factorized 1024-by-1024 grid. For this grid, the gating function only needs to predict two vectors of size 1024. This work also demonstrates that such layers can benefit Transformer models in the masked language modeling task.

However, these works require a centralized infrastructure for training. When the gating function picks appropriate experts for the input at hand, it must somehow find these experts across all nodes. In our scenario, even maintaining the dynamic "address book" of all active experts would be infeasible for any single participant.

## 2.4   Distributed Hash Tables

Fortunately, there is a way to implement bookkeeping in a decentralized system — the distributed hash table (DHT). This is a family of distributed data structures that store key-value pairs across multiple computers in a network. A single computer within such structure only needs to "know" $O(\log N)$ out of $N$ computers; at the same time it can look up any key with at most $O(\log N)$ requests to his peers. There are several DHT variants, but they all have common properties:

- **Decentralization:** nodes form and maintain DHT without any central coordination;
- **Scalability:** DHT can scale to millions of active nodes that are continually joining and leaving;
- **Fault tolerance:** a failure in one or a few nodes does not affect DHT integrity and availability;

A DHT-like protocol was first proposed in 1998 by [51] and popularized in early 2000s by four protocols: CAN [52], Chord [53], Pastry [54] and Tapestry [55]. By far, the most popular DHT variation is Kademlia [56] with numerous applications such as BitTorrent, I2P, and Ethereum. A more recent work [57] further improves theoretical performance for either lookup time or the number of connections; however, this version is less widespread due to being significantly harder to implement.

# 3   Learning@home

Our main idea is to use the existing properties of mixture-of-experts and distributed hash tables to work around the limitations of volunteer computing. We begin with a method for distributed training of MoE layers, then extend it to provide fault tolerance and decentralized bookkeeping.

## 3.1   Decentralized Mixture-of-Experts

The fundamental building block of our approach is Decentralized Mixture-of-Experts (DMoE) — a layer that contains multiple independent "expert" sub-networks distributed over a pool of workers. In addition to experts, each worker has a gating function: a lightweight sub-network that selects experts depending on the input. Similarly to regular mixture-of-experts, DMoE is a general-purpose layer that can process any input type by using the appropriate experts (e.g., convolutional or attentive).

Workers within the DMoE layer interact using Kademlia DHT protocol (Section 2.4). This DHT stores metadata, such as expert weights and worker status. Figure 2 explains DMoE inference:

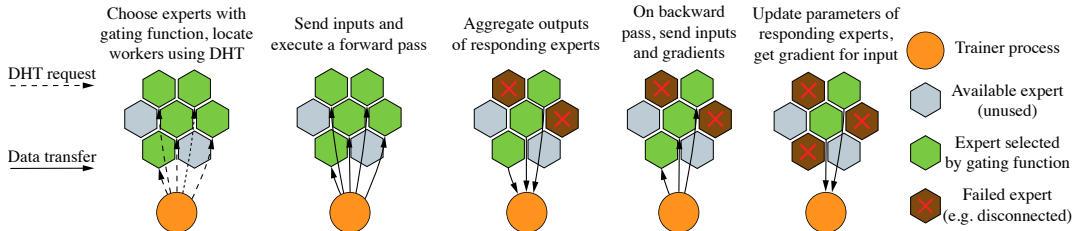

Figure 2: Forward and backward passes for Decentralized Mixture of Experts.

This procedure takes at most $O(k \log N)$ DHT queries to locate the chosen experts and $k$ direct interactions with these experts to do the actual processing. As long as $k \ll N$, we can increase the total number of experts without compromising the inference speed. Furthermore, we argue that DMoE layers automatically solve most of the issues that arise in the volunteer computing scenario.

**Fault tolerance.** If some of the $k$ chosen experts fail to respond due to a hardware or network error, DMoE can exclude those experts from averaging. The effect of such exclusion is similar to using Dropout [58] with regular mixture-of-experts. As a side effect, training DMoE on a faulty infrastructure will automatically adapt the mixture to the failure points of that infrastructure.

**Volunteer hardware.** Compute nodes can serve different numbers of experts based on their hardware capabilities. If one node leaves the network, another can take its place by retrieving the latest expert checkpoints from the DHT.

**Load balancing.** Mixture-of-experts layers can be regularized to balance the rate at which they select each expert in the mixture [47, 49]. Originally designed to improve MoE quality, this regularization has a side-effect of improving resource utilization by balancing computation load between workers.

**Asynchronous training.** Due to communication latency in distributed systems, a single input can take a long time to process. The traditional solution is to train asynchronously [37]. Instead of waiting for the results on one training batch, a worker can start processing the next batch right away. This approach can significantly improve hardware utilization at the cost of stale gradients.

Fortunately, Mixture-of-Experts accumulates staleness at a slower pace than regular neural networks. Only a small subset of all experts processes a single input; therefore, two individual inputs are likely to affect completely different experts. In that case, updating expert weights for the first input will not introduce staleness for the second one. We elaborate on this claim in Section 4.2.

### 3.2 Structured Gating Function

Since DMoE can use up to millions of experts, the gating function can no longer iterate over each expert in the mixture. Furthermore, the nodes in such a system are continually joining and leaving. Consequently, the expert selection procedure cannot rely on the availability of any individual node.

With this in mind, we propose a gating function inspired by product key layers [49]. First, we organize experts into a $d$-dimensional grid. Each expert $f$ is associated with a unique tuple of integers: $\mathrm{uid}(f) = (u_0, u_1, \ldots, u_{d-1}), u_i \in [0, M)$. The grid dimensions $d, M$ should be chosen to accommodate all experts with some level of redundancy. Having extra grid space allows DMoE to allocate additional experts midway through training if more volunteers join.

The gating function itself consists of $d$ linear layers $g_0, \ldots g_{d-1}$ and computes expert priority in an additive manner: $g(x, f) = \sum_{i=0}^{d-1} g_i(x)[u_i]$. Such a function only needs to predict $d$ vectors of size $M$, which makes it significantly easier to compute and send over the network. Furthermore, this gating function can choose top-$k$ highest-scoring experts in logarithmic time (see Appendix B, C). After choosing the appropriate experts, a worker should find their respective servers (in $O(k \log N)$ time using DHT) and pass the input vector for processing (see Figure 1). Once all the experts have finished processing, the worker aggregates expert outputs by weighted averaging:

$$\mathrm{DMoE}(x) = \sum_{f \in \mathrm{TopK}(x)} f(x) \frac{\exp\left(g(x, f)\right)}{\sum_{f' \in \mathrm{TopK}(x)} \exp\left(g(x, f')\right)} \ , \ \mathrm{TopK}(x) \text{ are } k \text{ best experts w.r.t. } g \quad (1)$$

If some of the chosen experts have crashed or taken too long to perform the computation, we can exclude them from averaging and renormalize weights so that they still add up to 1. Trained with this exclusion policy, DMoE will learn experts with overlapping specializations that are more resistant to individual node failure.

### 3.3 Training infrastructure

Finally, we describe Learning@home — a deep learning infrastructure that performs distributed training of large models on hardware provided by volunteers. Each worker runs three components:

- **Trainer** — forming batches and training;
- **Runtime** — inference and expert updates;
- **DHT Node** — bookkeeping and routing;

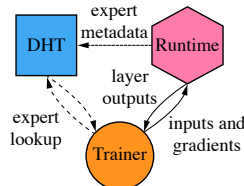

Figure 3: Learning@home components and their interaction.

**Trainer** generates batches and propagates them through the model. After forming a batch and converting it into an input vector, the trainer iterates over a sequence of DMoE layers and organizes forward and backward passes, as described in Sections 3.1 and 3.2. Learning@home fully embraces the asynchronous training paradigm, where a trainer can process hundreds of concurrent batches.

**Runtime** is responsible for expert inference and training. This is the only process that has access to participant's GPU device(s). Once all the experts are initialized, runtime listens to the incoming connections from trainers and handles two types of requests:

- **Forward**: given inputs, compute and return expert outputs on these inputs (no side-effects);
- **Backward**: given inputs and gradients of loss function w.r.t. outputs, return gradients w.r.t. inputs and *update expert parameters by gradient descent*.

Since trainers can operate under latency, the runtime is not required to process all requests right away. Instead, it aggregates requests into batches for better GPU utilization.

The runtime process relies on gradient checkpointing to avoid storing intermediate expert activations [59, 60]. This choice means that the expert $f_i(x)$ is called both during the forward and the backward passes. We elaborate on the role of gradient checkpointing in Appendix D.

**DHT Node.** The final component of Learning@home infrastructure is a DHT for bookkeeping. For simplicity, we use unmodified Kademlia protocol[7], leaving further investigation to future work.

Each runtime periodically announces its experts to the DHT, associating their identifiers with the address of that runtime and the current timestamp (details in Appendix C). Trainers can then use those entries to find the workers responsible for the chosen experts. In addition to timestamps, a runtime also regularly saves latest expert weights into the same DHT for persistence. The resulting infrastructure becomes elastic and fault-tolerant as long as it has enough active participants.

## 4 Experiments

The design of Learning@home was driven by two key assumptions: first, that MoE-based architectures can maintain high throughput under latency and second, that they can converge despite the presence of stale gradients. In this section we run several benchmarks in order to verify these assumptions. We intentionally focus on small-scale experiments to make them easier to reproduce and analyze. While solving practical vision and NLP problems is certainly our end goal, choosing a particular task would make it much harder to understand the general properties of our approach.

### 4.1 Model throughput

Our first benchmark evaluates the performance of asynchronous training schemes under latency. We quantify this with training throughput, i.e., the number of training batches processed per second. To emulate the distributed training environment, we create a model from a large number of identical blocks distributed evenly across 4 NVIDIA GTX 1080 GPUs. We simulate network latency by adding an artificial delay after computation of each block. The delay time is sampled from the exponential distribution, which was shown to model latency well [61].

Since our model size exceeds the memory limits of a single consumer GPU, the only mainstream paradigm that can compete with Learning@home is model parallel training. We also report the "upper bound" on training throughput by running the same computations with no network delays in a model parallel regime with pipelining similar to [4]. For Learning@home, we use 64 trainer processes to send requests to the runtime processes[8].

To measure the effect on blocks with different computation to communication ratio, we evaluate two popular block architectures. The first architecture is composed of 224 feed-forward blocks, each having hidden dimensions of $1024 \rightarrow 4096 \rightarrow 4096 \rightarrow 1024$ with layer normalization and ReLU activations in between. These blocks are treated as separate "experts" and process batches of size 2048. The second architecture consists of 224 BERT-like Transformer blocks [7] with hidden dimension 1024 and GELU activations [62] applied to sequences of length 512 with batch size 4.

With this setup in mind, we can measure the throughput of the entire model as the time it takes to process 10 batches and dividing it by the total number of processed examples. These experiments were repeated 5 times for all methods to measure the mean and standard deviation of throughput.

Figure 4 demonstrates that even with delay times approaching 200ms the asynchronous scheduler we have implemented as part of Learning@home maintains nearly the same throughput. In turn, model-parallel training throughput quickly degrades under latency, which is not surprising as it was not designed with slow communication in mind.

To verify the validity of our conclusions, we have conducted similar experiments on cloud GPU instances in different regions. This allows us to measure performance in a non-simulated scenario closer to the desired area of application. In particular, we rented 3 instances with Tesla K80 hosted in West US, East US, and West Europe with average network latency of $92.49 \pm 32.42$ ms. The throughput values in Table 2 are similar to results for simulated latencies (Figure 4).

Finally, we tested the scalability of our infrastructure by deploying DHT nodes in the same cloud regions and measuring the latency of beam search (batch size 64, see Appendix C). Finding top-4 experts took $317 \pm 58$ms for 100 nodes, $528 \pm 127$ms for 1,000 nodes and $764 \pm 106$ms for 10,000 DHT nodes.

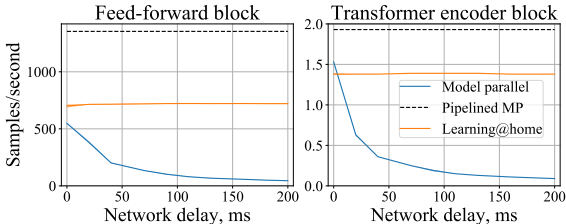

| Approach | Feed-forward | Transformer encoder |
|---|---|---|
| Model parallel | $7.23 \pm 0.06$ | $0.01 \pm 0.001$ |
| Learning@home | $300.8 \pm 15.9$ | $0.68 \pm 0.01$ |

Table 2: Throughput (samples/s) for 3 cloud K80 in East US, West US and West Europe.

Figure 4: Throughput with simulated latency.

### 4.2 Convergence

Our second experiment aims to verify the robustness of DMoE to delayed updates. For this goal, we choose one of the simpler tasks in deep learning, namely the MNIST digit recognition dataset [63], and compare convergence rates under varying network latency. All modern architectures can reliably solve this task, making it easier for us to isolate the effect of gradient staleness.

We evaluate four models: a traditional feed-forward model and three DMoE variations with different numbers of experts. The feed-forward network (FFN) consists of 4 stacked feed-forward blocks. Each block architecture is same as described in Section 4.1, but with half as many hidden units. In turn, its DMoE counterparts have four DMoE layers, each composed of blocks with 1/4 of the FFN size. Both DMoE-based models use only 4 experts at a time regardless of their total number, hence being computationally equivalent to the FFN baseline.

We train all models asynchronously in high-latency and low-latency scenarios, using the same distribution for delay. In the high-latency scenario, each of 64 workers is delayed for 1 second on average while processing a batch. This corresponds to 125ms for each forward and backward pass through DMoE. For low latency emulation, we use 16 workers and 100ms average delay. The third experiment simulates node failure: each expert does not respond to a request with probability 0.1.

The results are presented in Figure 5; as expected, the plots demonstrate that the higher latency scenario is more difficult for all models. However, the degree to which it affects the performance of DMoE architectures is much lower, especially for the largest of mixtures.

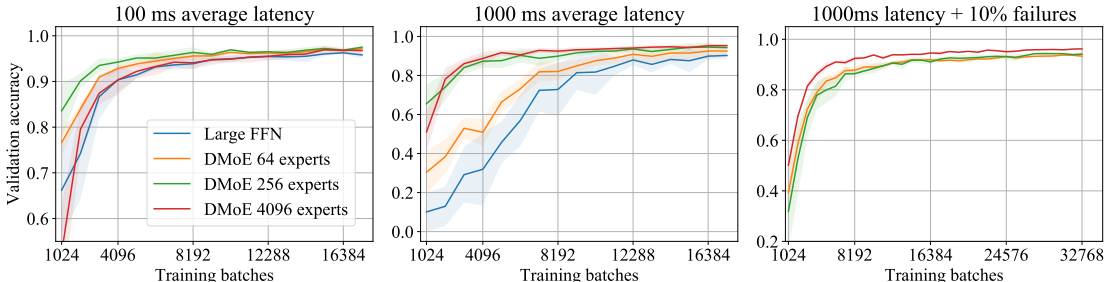

Figure 5: Convergence plots for feedforward models with different network latencies and failure rates. Pale areas on depict unbiased standard deviations over 5 runs.

### 4.3 Language models

The third and final benchmark is neural language modeling. Specifically, we train Transformer-XL [64] on the WikiText-2 [65] dataset. Both baseline and DMoE models use official recommended parameters with additional regularization proposed in [66].

The `base` model contains 16 Transformer layers with the hidden size of 400 and 900 units in the feedforward layer. We also train a `small` baseline model with 200 hidden and 450 feedforward units. Our DMoE Transformer uses 256 experts split evenly between 16 layers. Each expert is a Transformer layer with the same dimensions as layers of the `small` baseline model. The DMoE layers route to top-4 experts, making our model roughly equivalent to `base` in terms of FLOPs per sample. Similarly to Section 4.2, we train DMoE with 32 trainers (batch size 1 each), 1000ms average latency, and 10% failure rate.

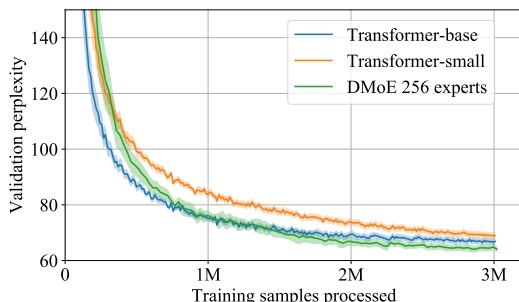

Figure 6: Convergence plots for Transformer language models on the WikiText-2 dataset. Pale areas on depict unbiased standard deviations over 5 runs.

The results depicted in Figure 6 demonstrate a similar pattern to what was previously observed on feedforward networks. Curiously enough, we found that in this specific scenario the 10% failure rate has a positive effect on the DMoE performance. We attribute this effect to a form of dropout regularization that prevents our model from overfitting the limited training data.

## 5 Conclusion

The main purpose of this study is to convey the idea that one *can* train large neural networks on unreliable hardware. We propose a specialized layer and training infrastructure designed to meet the requirements of volunteer computing over the Internet. The preliminary experiments demonstrate that Learning@home can scale to thousands of nodes and successfully train popular model archetypes despite network latency and node failures.

We believe that decentralized deep learning will change the way we think about training neural networks. Instead of running isolated experiments, researchers and practitioners will be able to join forces and solve the biggest problems together. Instead of being confined to a single supercomputer, our models will naturally grow in capacity as more people and organizations around the world join in. We expand on the ramifications of deep learning decentralization in the broader impact statement.

However, reaching the full potential of this idea requires expertise not only in deep learning, but also information security, distributed systems, crowdsourcing and many other areas. We believe that this monumental task is best solved through scientific collaboration. To that end, we will continue to develop Learning@home as a public open-source project[9].

### Acknowledgements and funding

We would like to thank Artem Babenko and Vladimir Aliev for their invaluable assistance in both brainstorming and proofreading the final paper. We are also grateful to anonymous reviewers for their helpful suggestions on improving the presentation of the paper. Max Ryabinin was supported by Yandex and National Research University Higher School of Economics.

## Broader Impact

The approach proposed in this work is only a prototype with limited direct consequences, but the long-term goal of training huge models with volunteer computing can have a lasting effect on both the research community and the general public.

### Funding bias vs crowdsourcing bias

The main positive outcome we pursue is to let researchers harness volunteer computing and train models on the scale currently available only to large corporations. Ideally, a deep learning researcher with a promising idea will be able to amass the computation needed to realize this idea by involving volunteers. However, the project's appeal for volunteers depends on many factors such as subject area, current societal trends, and even researcher's personality.

For example, a project about teaching agents to play games [38] or fighting global pandemics [67] is likely to attract more resources than deep learning applied to soil science. In essence, volunteer computing is biased towards exciting or socially relevant research the same way as traditional HPC is biased towards the interests of those who fund it.

### Alternative use and misuse

The proposed technology can be used with different economic models. If a deep learning system is immediately useful (e.g. for machine translation, information retrieval, etc), the participants could use it for their needs based on their contributions to training. This can take many forms: several labs combining their hardware and training larger models; a web-service that lets people contribute their compute instead of using ads/subscriptions; or simply a framework that someone can use to run distributed training across two or more datacenters.

Unfortunately, this also allows several opportunities for malicious use. If a machine is hacked, the attacker can use its compute unnoticed by the machine owner — much the same way that botnets are currently used to mine cryptocurrencies. Furthermore, due to decentalized nature even legitimate Learning@home projects can be hijacked by hackers.

### Security

Using crowdsourced hardware makes Learning@home susceptible to attacks from malicious participants. There are multiple attack vectors already known in P2P community: denial of service attacks, Sybil attacks, Eclipse attacks and more [68, 69, 70, 71]. Fortunately, there are variations of the DHT protocol that make it resistant to said attacks: if a reader wishes to learn more about DHT security, we recommend starting with [68].

Another source of vulnerability stems from the sequential nature of neural networks. If a single expert were to return incorrect (e.g. NaN) outputs or gradients, it could compromise the outputs of the entire network and even poison adjacent nodes through backpropagation. Recent studies expose similar attack patterns on federated learning systems [72, 73].

The redundant nature of mixture-of-experts layers provides some degree of resistance against those attacks. A single malicious expert will only affect a small fraction of inputs that pass through this specific expert. Furthermore, a trainer with access to predictions from multiple experts could provide a higher degree of robustness by using statistical techniques (e.g., by ignoring outlier gradients). However, such techniques need to be carefully designed so as not to introduce harmful side effects.

### The burden on the network

Finally, we would like to point out the potential harm that our approach can do to network infrastructure. The experiments we ran in Section 4.1 saturate with the bandwidth of $100 - 200$Mbps, most of which is tensors passed between experts and trainers.

This coincides with the typical home internet speed available in major cities of developed countries. However, not all ISPs design their infrastructure for users who always use up all their bandwidth. If too many Learning@home participants are located in one LAN or MAN, it can cause congestion or even failures in the network infrastructure.

Similar situations frequently took place in late 2000s due to growing popularity of BitTorrent for file sharing. Fortunately, the network infrastructure is continually improving, which leads us to believe that this problem will eventually be solved. Until then, we describe several ways to reduce network load of Learning@home in Appendix E.

## Footnotes

[3]`https://github.com/mryab/learning-at-home`

[4]In January 2019, Folding@home reported 146,091 teraflops; in November 2019, the top-1 supercomputer "Summit" reported 148,600 teraflops; see `top500.org/lists/2019/11` .

[5]Typical specifications: 2–8 CPU cores, 4–16GB RAM, and a single customer-grade GPU with 2–12GB of memory and 4–14 float32 TFLOPS (based on `https://pcpartpicker.com` and `https://techpowerup.com`)

[6]We assume 20–250ms latency and 100Mbps symmetric bandwidth, $0.33\%$ packet loss based on [23, 24]

[7]In particular, publicly available Kademlia implementation from `github.com/bmuller/kademlia`

[8]See the full setup: `https://github.com/mryab/learning-at-home#running-the-experiments`

[9] `https://learning-at-home.github.io`

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
