[Supplementary Material]

# Towards Crowdsourced Training of Large Neural Networks using Decentralized Mixture-of-Experts Supplementary Material

**Max Ryabinin**
Yandex
National Research University
Higher School of Economics
mryabinin@hse.ru

**Anton Gusev**
Independent
uartman@mail.ru

## A   Cost and performance estimate of $2500 desktop PCs

According to several PC building websites (https://pcpartpicker.com, https://newegg.com), most popular $2250–2750 desktops are equipped with RTX 2080/2080Ti or GTX 1080Ti GPU. These GPUs are 50–80% as fast as Tesla V100 for deep learning [1]. As a rough estimate, the combined throughput of 10,000 desktops is 8–15 times that of server pod with 512 V100 GPUs.

## B   A primer on Distributed Hash Tables

On a high level, DHT is a dictionary that can be accessed by every participant. Each key-value pair is stored on a small subset of peers determined by the hash function of the key.

- Each participant has a unique identifier (ID) that is sampled uniformly from the space possible outputs of the hash function.
- When storing a $(key, \ value)$ pair, one should search for $k$ peers whose IDs are closest to $\mathrm{hash}(key)$. Then, request each of these $k$ peers to store the $(key, \ value)$ pair.
- When retrieving a value for a key, one should compute $\mathrm{hash}(key)$, search for peers with IDs similar to that hash value and request value from those peers.

Specific DHT variants such as Chord [2] or Kademlia [3] employ different hash types and different algorithms for finding nearest peers. For instance, Kademlia DHT selects nearest peers based on the XOR distance function: $d(x, y) = \mathrm{int}(x \oplus y)$.

Each participant is directly aware of only a small subset of DHT peers. When storing or retrieving a key, the participant requests additional peers from its neighbors in a semi-greedy search, minimizing XOR distance until it finds $k$ nearest peers. In Kademlia, nodes form a special navigable graph structure that lets them find nearest peers in at most $O(k + \log_2 N)$ requests to other DHT peers, where $N$ is the total number of participants.

## C   Finding best experts across the DHT

Recall that the gating function is defined as

$$g(x, f) = \sum_{i=0}^{d-1} g_i(x)[u_i],$$

where $g_0, \ldots g_{d-1}$ are linear layers, $u_i$ is the $i$-th component of the expert unique identifier $\mathrm{uid}(f)$, and $[k]$ takes $k$-th component of a vector. Our objective is to find $k$ experts with largest $g(x, \cdot)$. In

a centralized setting, one can find $k$ largest scores from each linear layer $g_i$ using the algorithm described in [4].

Unfortunately, in our case not all combinations of indices correspond to valid experts. Therefore, we developed a specialized beam search algorithm similar to the one used in machine translation. The core idea is to start with top-$k$ indices along the first grid dimension and add one dimension at a time.

In order for this algorithm to work, participants maintain the following information on the DHT:

- For every expert UID, store its server address and the timestamp;
- For every prefix in expert UID, store all suffixes corresponding to active experts and the timestamp.

For instance, if there are 6 experts: "ffn.1.3", "ffn.2.1", "ffn.2.2", "ffn.2.6" and "ffn.3.2" and "ffn.3.5"; the DHT will contain the following information:

| Key | ffn.1.* | ffn.2.* | ffn.3.* | ffn.1.3 | ffn.2.1 | ffn.2.2 | ffn.2.6 | ffn.3.2 | ffn.3.5 |
|---|---|---|---|---|---|---|---|---|---|
| Value | [3],$t_1$ | [1, 2, 6],$t_2$ | [2, 5],$t_3$ | [Address of a server that hosts the given expert] | | | | | |

Figure 1: DHT keys and values for 6 experts defined above, t corresponds to last update timestamp.

For higher grid dimensions, we store similar information for every grid prefix. For instance, an expert with UID "transformer.10.20.30" will affect 3 keys: "transformer.10.*", "transformer.10.20.*" and "transformer.10.20.30". Each prefix key stores at most as many values as there are indices in the next grid dimension, typically 100 or 256.

With this data structure, DMoE can use beam search to select the best experts. Algorithm 1 starts from the leftmost dimension of the grid and processes one dimension at each step. The worst case complexity of this algorithm is $O(dk \log N)$ from $O(dk)$ lookups to the DHT.

---

**Algorithm 1** SelectExperts

---

   **Input:** $x, k, d, M, (g_0, \ldots, g_{d-1})$
   beam := $[0, 1, ..., M - 1]$               // all 1-prefixes
   scores := $[g_0(x, 0)...g_0(x, M - 1)]$         // initial scores
   // select $k$ best starting points
   beam, scores := TopK(beam, scores, k)
   **for** $i \in [1, \ldots, d - 1]$ **do**
      // expand all candidates in beam
      new_beam, new_scores := $[\,], [\,]$
      **for** prefix, score $\in$ beam, scores **do**
         **for** $j \in$ ActiveSuffixes(prefix) **do**
            new_beam.add(prefix$\bigoplus[j]$) // concat
            new_scores.add(score $+g_i(x, j)$)
         **end for**
      **end for**
      // select at most $k$ best prefixes
      beam, scores := TopK(new_beam, new_scores, k)
   **end for**
   **Return** beam

---

The TopK function simply sorts the inputs by score and returns $k$ inputs with highest scores. In turn, the ActiveSuffixes function queries the DHT for a given prefix and returns a set of all active suffixes as described above. Assuming that servers re-publish their experts every $t$ seconds, the function can simply check whether the timestamp for a given prefix is less than $t$ seconds old.

# D   On gradient checkpointing in Learning@home

In general, gradient checkpointing increases computation per training batch by approximately 1/3, but allows training larger models with the same GPU memory. More importantly, in our scenario

checkpointing also removes the need to store intermediate activations. In our experiments, this has led to both significantly higher training throughput and a smaller memory footprint.

Without gradient checkpointing, we would have to store intermediate activations in memory. Since the GPU can only fit a few batches at a time, it quickly runs out of memory and is forced to wait for the backward pass. For Transformer layers (see Figure 4, top), this results in approximately 9 times less throughput at 100ms latency.

## E    Reducing the network load

One way to reduce the communication load is to convert tensors to a lower precision before transfer. Prior work in this area suggests that distributed training works even when communicating with 8-bit precision tensors [5, 6]. Many popular architectures, including Transformers, can train entirely in that precision mode [7]. Consequently, low precision communication appears as a logical way of reducing communication requirements.

In addition, the deep learning architectures discussed in this work rely on backpropagation for training. With the advancement of optimization methods allowing nearly independent layer-wise training [8, 9, 10], it might be even more suitable to use these techniques for asynchronous training with fewer restrictions on the architectures being used.

Another solution is to use experts that have a higher capacity to input size ratio. The architectures used in Section 4.1 are already somewhat biased in that direction, but they are far from optimal.