[Reviews · NeurIPS 2020]

Review 1

Summary and Contributions: This work proposes a methodology to train massive neural networks over distributed, heterogeneous and unreliable hardware. To do this, it proposes a Decentralized Mixture-of-Expert (DMoE) layer along with a training framework, which extends DMoE models over such training infrastructures. However, though this sparse architecture is generally amenable to the demands of the application, the authors note that MoE models scale poorly over flat layers of experts. They therefore propose a new gating function via product key layers and a distributed hash table. I think this is a very promising direction and I strongly encourage the authors to continue this research and build the training infrastructure. However, as of this paper, it reads more as a proposal document and is light on empirical results.

Strengths: * This paper addresses an important problem for scaling and suggests a path forward with the compute available with consumer-grade PCs connected via the internet. This is an ambitious and an important problem. * The early empirical results demonstrate the efficacy of Learning@home versus model-parallel systems and the graceful degradation of MoE layers with high latency is encouraging.

Weaknesses: * Measuring the performance and tradeoffs of this system against established performance benchmarks would be necessary. This work supports that the network converges, but it is not clear what performance trade-off exists with this new architecture and asynchronous updates. For instance, does this architecture and training regime facilitate competitive natural language processing or computer vision performance? The only performance measure currently benchmarked is that this architecture/system trains MNIST models that exceed 95%+ validation accuracy. * The related work extends nearly three pages. The experimental section, arguably the most important for an empirical and engineering research, is squeezed into a page. This is well-motivated work, but more empirical testing and further details of the infrastructure is needed. * There is of course no demand that the authors to demonstrate the full potential of an idea ("However, reaching the full potential of this idea is a monumental task well outside the restrictions of one publication."), but I'd generally recommend withholding potential subjective measures of what is or is not within the scope of a publication.

Correctness: All claims and methods appear reasonable and correct

Clarity: * The approach is clearly delineated, but I might suggest to the authors a less conversational tone in scientific writing (L49: “all that power”, L50: “way slower”). This is of course a matter of style, but some readers may find it distracting. * As one recommendation, aim to strengthen your conclusion. You’ve laid out an important and interesting research direction, but the conclusion does this paper a disservice. Don’t squeeze it in at the bottom as an after-thought. * Recommend a separate dedicated section for Training Infrastructure, one of the primary contributions. * What is a block architecture? The model for the MNIST digit recognition is unclear.

Relation to Prior Work: Volunteer computing over internet-connected serves and devices is a widespread technique, but the specific application of a Mixture-of-Expert layer is novel as far as I am aware.

Reproducibility: Yes

Additional Feedback: * Why does Learning@home have a higher throughput in examples/second than the model-parallel system at 0ms network delay? * How do existing deep learning frameworks fail to support DMoE architecture? ============= Post review: Thank you for the additional experiments and revisions. I have raised my score accordingly.


Review 2

Summary and Contributions: The work introduces the merger of ideas from distributed hashtables and mixture of experts with the goal of enabling volunteer computing in large scale ML.

Strengths: The paper is very well written. I did not find a single typo or odd formulation. This is very much appreciated. The core idea is solid. I really like the thinking down these lines, and how we can make ML more accessible to more researchers. Using spare CPU and GPU cycles is a great way to do that.

Weaknesses: The main motivation in the paper is a bit simplistic and inwards-facing on the ML community. The impact statement expands on this and the challenges of what work *doesn't* get done because only work related to the interests of large organizations can get done. I'd advise the authors to move some of that thinking into the main paper. The evaluation is weak, and the authors admit as much. Without deployment, it is near impossible to say whether this approach can work in practice. I am also skeptical of some of the assumptions (e.g. symmetrical 100MBit/s connections), but have equally no empirical basis for that skepticism.

Correctness: Very hard to say. The arguments laid out are clear and believable. However, this being a distributed systems proposal on volunteer hardware, I have no way of knowing whether it actually works.

Clarity: Yes, very much so. Thank you!

Relation to Prior Work: Yes. However, I would encourage to spend a few more lines on DHTs and their core mechanisms. The NeurIPS audience is generally not well versed in systems.

Reproducibility: Yes

Additional Feedback: I wonder whether this would be better as a demo, followed by a deployment and then a paper. This would be infinitely more exciting to read if there was a section on practical experiences with the proposed system. Comments added after author feedback ============================== Thank you for making the time to address my concerns. My main concern remains that it is hard to know how a P2P system will perform until it is deployed. Hence, I cannot raise my score.


Review 3

Summary and Contributions: This paper raises the question that what would be the solution if the model and data size are 1000x than the current ones. It points out that volunteer computing could be the right direction by proposing a framework/system to train large neural networks on volunteer hardware. That hardware is usually poor devices like regular PCs. Finally, it evaluates the prototype on several real tasks.

Strengths: 1. Although making use of the poor individual devices for distributed deep learning model training is not a new idea, building a working prototype or a system is a real contribution. 2. Also, although the idea of moe is there for a while but building this decentralized moe in volunteer computing can be useful to the community from the system perspective. 3. The paper is well-written. Specifically I enjoy the flow (intro is very interesting and I appreciate the efforts in related work to make the paper more fluent) and every point it is making is crystal clear. In general, the major reason why I like this paper is that 1) machine learning or deep learning at scale will someday absolutely become a major challenge and this paper provides a good angle to look at this problem 2) there are many ideas/advances in DL everyday but not all of them providing a working system. But also because of this, I didn't give a high score. Details in weaknesses. --------------UPDATE-------------- To clarify what the author feedback states: I'm not claiming the distributed training is not a new idea, but I'm saying Mixture of experts (MOE) has been out and show advantages in large-scale training for a while.

Weaknesses: 1. To me the major drawback of the paper is its evaluation. After reading the inspiring introduction. I was expecting a billion trillion scale evaluation and surprising number in the end because that was the whole motivation. It is very disappointing to see this kind of poor evaluations in a paper with such a good start. 2. While I understand the importance of first two sections, the intro and background section has over four pages. I recommend shrinking these sections and adding more evaluations. I see authors state in the beginning of the evaluation that it is toy for easy reproducibility. If authors could provide further large scale experiments/evaluations, I will raise my score. Otherwise it is hard to convince audiences that this paper is solving a trillion scale problem but the evaluation is on MNIST. ---------------Update------------------- I'm raising my score because the author has shown a relative larger scale deployment with transformer-xl and wiki2. (The dataset is not that large but much better than MNIST) But the main reason is still, I think this direction needs some support and encouragement. I hope that, despite this paper got accepted or not, the authors can keep up such practical research and friendly code-base.

Correctness: Yes.

Clarity: Yes.

Relation to Prior Work: Yes.

Reproducibility: Yes

Additional Feedback:

[Author Response · NeurIPS 2020]

We thank the reviewers for their feedback. A common motive in all reviews is that decentralized training is a promising research direction that can solve major challenges of model scaling and accessibility. Up until now, this area has not been actively discussed in the academic community. Hence, we believe that the publication of Learning@home will impact diverse application fields and invite broad academic expertise that a single research team cannot have.

**(R2) "The evaluation is weak, and the authors admit as much."**
**(R4) "If authors could provide further large scale experiments/evaluations, I will raise my score."**
**(R1) "not clear what performance trade-off exists with this new architecture and asynchronous updates"**

To better support our claims, we conduct additional experiments on the language modeling task. Specifically, we train Transformer-XL [1] on the WikiText-2 dataset. Both baseline and DMoE models use official recommended parameters with regularization implemented in [2]. The `base` model contains 16 Transformer layers with hidden size of 400 and 900 units in the feedforward layer. We also train a `small` baseline model with 200 hidden and 450 feedforward units. Our DMoE Transformer uses 256 experts split evenly between 16 layers. Each expert is a Transformer layer with the same dimensions as layers of `small` baseline model. The DMoE layers route to top-4 experts, making our model equivalent to `small` in terms of FLOPs. As in Section 4.2, we train DMoE with 32 trainers (batch size 1 each), 1000ms average latency, and 10% failure rate. Figure 1 shows that DMoE outperforms the baseline with the same compute budget.

Figure 1: Results on WikiText-2.

**(R1) "does ... facilitate competitive natural language processing or computer vision performance?"**
**(R2) "The main motivation in the paper is a bit simplistic and inwards-facing on the ML community."**

Fortunately, practical performance benefits of MoE-based models for natural language processing were demonstrated in a concurrent preprint [3]. This study reports training MoE-Transformer with 600B parameters for multilingual machine translation using 2048 TPUv3 accelerators with gains of up to +13.5 BLEU (page 16, Figure 6).

In turn, Learning@home infrastructure provides a way of training such models using volunteer hardware instead of a TPU cluster. We understand that this claim would be better supported by a training campaign with thousands of volunteers. Due to the restrictions of anonymity, the best evidence we can provide is that Learning@home runs reliably with 10,000 CPU-only participants (L308) and models with a memory footprint of up to 192Gb (Section 4.2).

**(R1)(R2)(R4) Feedback on paper clarity and presentation.**

While all reviewers agree that the paper is well-written, they suggest similar improvements to the presentation. We will incorporate these suggestions in the final version of the paper:

- (R1, R4) Reduce the length of the "Related Work" section to free up space for additional experiments;
- (R2) In turn, create "Additional Related Work" section in supplementary materials for further details, including a more detailed discussion on the background and inner mechanisms of Distributed Hash Tables.
- (R1, R2, R4) Use the freed space to report WikiText-2 experiments (see above) and expand the conclusion.

**(R1) "Volunteer computing over internet-connected servers and devices is a widespread technique"**
**(R4) "Making use of the poor individual devices for distributed deep learning model training is not a new idea"**

Though this is technically correct, there is only one study (Kijsipongse et al, 2018) that applies volunteer computing to general deep learning. Their approach requires that the model fits in the GPU memory of the weakest participant. Other projects share this drawback and only operate on niche tasks such as playing board games (see L130-138).

**(R1) "less conversational tone in scientific writing (L49: "all that power", L50: "way slower")"**

We agree that most colloquial phrases can be replaced with more formal language without reducing text clarity.

**(R1) "What is a block architecture?"** The feedforward block used in Section 4.2 is the same as described in Section 4.1 (L290), but with half as many units in every dimension. We will clarify the description to avoid reader confusion.

**(R1) "How do existing deep learning frameworks fail to support DMoE architecture?"** Existing DL frameworks (e.g. TensorFlow/PyTorch) support mechanisms for model-parallel training, but these mechanisms can't recover from node failures. Other tools such as TorchElastic ensure fault tolerance but are incompatible with model-parallel training.

**(R1) "Why does Learning@home have a higher throughput ... at 0ms network delay?"** Even without network delay, the batch processing time will still fluctuate due to device specifics, leading to delays in model-parallel training.

**(R2) On 100Mb/s symmetric bandwidth assumption.** We will add further justification of this assumption based on Speedtest global index [4]. To summarize, the symmetric bandwidth in top-20 countries is generally in the 90–200Mb/s range, and the global average is steadily increasing.

[1] Z. Dai et al. "Transformer-XL: Attentive Language Models beyond a Fixed-Length Context." ACL 2019.
[2] https://github.com/TimDettmers/transformer-xl
[3] D. Lepikhin et al. "Gshard: Scaling giant models with conditional computation and automatic sharding." arXiv:2006.16668 (2020).
[4] Speedtest Global Index for Fixed Broadband https://www.speedtest.net/global-index (11.08.2020)


[Meta-Review · NeurIPS 2020]

This paper focuses on the use of "citizen science" to train large neural networks. An algorithm is proposed that is fault-tolerant to missing/slow/unreliable nodes, and some preliminary experiments are carried out to demonstrate its utility. The reviewers initially suggested that the experiments were limited, but after rebuttal were convinced that the paper is worth publishing in its current form.